# Poorer subjective mental health among girls: Artefact or real? Examining whether interpretations of what shapes mental health vary by sex

Susan P. Phillips[1], Fiona Costello[1], Naomi Gazendam[1], Afshin Vafaei[2]*

1 Department of Family Medicine, Queen's University, Kingston, ON, Canada, 2 School of Health Studies, Western University, London, ON, Canada

* avafaei2@uwo.ca

**Data Availability Statement:** Data will be available upon reasonable request to the corresponding author. Requests can also be sent to Mary Martin, Research Manager, Department of Family

## Abstract

### Background

Despite reporting poorer self-rated mental health (SRMH) than boys, girls exhibit greater resilience and academic achievement, and less risk taking or death by suicide. Might this apparent paradox be an artefact arising from girls' and boys' different interpretations of the meaning of SRMH? We examined whether the indicator, SRMH, had a different meaning for girls and boys.

### Methods

In 2021–2, we circulated social media invitations for youth age 13–18 to complete an online survey about their mental health, and which of 26 individual and social circumstances shaped that rating. All data were submitted anonymously with no link to IP addresses. After comparing weightings for each characteristic, factor analyses identified domains for the whole group and for girls and boys.

### Results

Poor SRMH was reported by 47% of 506 girls and 27.8% of 216 boys. In general, circumstances considered important to this rating were similar for all, although boys focussed more on sense of identity, self-confidence, physical well-being, exercise, foods eaten and screen time, while girls paid more attention to having a boyfriend or girlfriend, comparisons with peers, and school performance. With factor analysis and common to boys and girls, domains of resilience, behavior/community, family, relationships with peers and future vision emerged. Girls' poorer SRMH did not arise from a more expansive interpretation of mental health. Instead, it may reflect perceived or real disadvantages in individual or social circumstances. Alternatively, girls' known greater resilience may propel lower SRMH which they use intuitively to motivate future achievement and avoid the complacency of thinking that 'all is well'.

Medicine, Centre for Studies in Primary Care, Queen's University at mary.martin@queensu.ca who acts as the institutional representative for this study.

**Funding:** Two authors (NG and FC) were supported by a summer studentship from a CIHR grant. (161787).

**Competing interests:** None of the authors declared any competing interest.

**Abbreviations:** SRMH, self-rated mental health; QR code, quick response code; CI, confidence interval; KMO, the Kaiser–Meyer–Olkin measure for sampling adequacy.

## Conclusions

The relative similarity of attributes considered before rating one's mental health suggests validity of this subjective measure among girls and boys.

## Background

Are consistent poorer ratings of mental health among girls relative to boys trustworthy? These ratings need to be juxtaposed with objective measures of girls' greater academic achievement, less risk-taking behavior, and lower suicide rates [1–4]. The paradox of girls' apparent poorer mental health alongside greater well-being is unexplained. We will explore whether the frameworks, circumstances and characteristics considered when subjectively rating mental health differ for girls and boys. If so, then SRMH could be measuring different constructs in each group and would, as a result, differentially reflect actual mental health. This matters for several reasons. If SRMH rests on girls' and boys' different interpretations of what constitutes good or poor mental health, it will not be a homogeneous measure and thus will be invalid [1, 5]. Statistical findings of a growing mental health crisis among youth, and especially girls are shaping health and education policies (e.g. calls for more mental health professionals, return to school plans post-pandemic, or 'accommodations' in higher education) [6–9]. If these interventions respond to unreliable information they risk being misdirected and might actually undermine rather than foster strengths and resilience.

Whether self or media reported, mental health of adolescents and particularly girls is increasingly said to be poor and declining [7, 10]. Administrative data capturing clinical diagnoses of anxiety and depression over two decades reveal a similar albeit weaker trend. In Canada, from 1997–2017 the rate of physician diagnosed anxiety and depression among girls exceeded that for boys, with small increments for all beginning in 2012 [11]. Subjective indicators are more compelling. In numerous, high-income countries, girls consistently and increasingly report greater anxiety, depression, stress, and psychological distress than do boys [12–16].

Apparent poorer mental health among girls exists alongside their greater (relative to boys) adaptability, resilience, academic achievement, self-control and less frequent risk-taking [17]. By resilience we mean positive adaptation in the face of adversity [18]. How might the inverse relationship, seen in girls, between mental health and these indications of well-being be reconciled? This is not a question that has been researched. In particular, there has been no consideration of whether this paradox could arise from gender divergence embedded in youth's interpretation of the measure, SRMH.

What, then, is gender and how is it different from sex? We define sex as a set of biological attributes based on chromosomes, gene expression, hormones, and reproductive anatomy, and gender as the way life is shaped by socially constructed roles, behaviors, and expressions ascribed on the basis of sex [19]. We recognize the importance of the psychological and material in constituting the gendered person [3]. When we use the term, gender, we acknowledge that sex and gender are interconnected in how lives are lived, and in shaping and rating health. Finally, we differentiate gender, a social determinant of health, from the individual-level characteristic of gender identity.

To identify what is actually being measured when youth are asked to rate their mental health it would be useful to explore differences in meanings attributed to this measure [20]. For example, girls' apparent poorer SRMH might arise from gender differences in

interpretations, approaches, and expansiveness with which the question is viewed, or from greater willingness to examine and name weaknesses. Previous research has demonstrated that adolescent girls are more alert to stressors and describe lower life satisfaction than do boys [21, 22]. Girls' internalization of reactions to lived problems, and higher expectations for themselves may produce anxiety. In contrast, boys tend to focus more on external circumstances such as life events that they cannot control and from which they, therefore, can absolve themselves of responsibility [16]. Although stigma around mental health is decreasing overall, girls' apparent introspection and reflection could heighten their insights about and willingness to name weaknesses [23]. In contrast, less reflective and more externalized traditional male responses could translate into a subjective rating of 'I am fine'. In keeping with resilience theory, it may be that more self-reflective youth and those with greater self-mastery and control hesitate to rate their MH highly, using a lower rating to push themselves to achieve [24].

To the best of our knowledge there have been no attempts to tease apart whether attributing different meanings to subjective measures of mental health underpin gender disparities in responses. Might the internalizing and externalizing described above lead each group (boys, girls) to weigh aspects of their lives (social supports, family relationships, etc.) differently? For example, if a girl considered family discord as a precursor of SRMH and her twin brother did not, despite similar lived realities their subjective mental health rating might be quite different. The girl and boy would, in effect, be answering different questions when asked to rate their mental health. A single indicator used for a complex health outcome such as mental health may in fact measure different constructs for different groups. This will possibly induce a measurement validity issue. Our aim was to determine what personal and social characteristics or experiences youth consider when making subjective mental health ratings, and whether these vary with for boys and girls. Such variation could partially explain girls' apparent SRMH disadvantage in the face of their greater academic and general well-being.

## Methods

This observational study was conducted via an online survey of > 700 youth age 13–18. We examined descriptive data, and then undertook factor analysis to identify latent factors that youth consider in rating their mental health.

### Data collection, study population, and the survey instrument

Data were collected from August 2021 to February 2022 via a 10-minute online survey created using the Qualtrics platform. To be included in this study participants had to be 13–18 years old and proficient in reading English. Although there were no geographic restrictions on participation, we specifically targeted Canadian youth in advertising.

This was a sample of convenience. Posters describing the survey were displayed (with school consent) in larger Canadian cities including Toronto and Vancouver. Additionally, recruitment occurred in Kingston, ON via posters on the Queen's University campus (i.e., residences, athletics centre, libraries), Kingston public libraries, cafés, community centres and Kingston's Limestone District School Board. These posters included a QR code to bring participants to the online questionnaire. Invitations to participate were also circulated electronically via social media. Specific invitations were sent to a small number of youth groups. The sample size used for analysis was 765, including 216 boys, 506 girls and 43 youth self-identified as non-gender binary.

Before distribution the survey was pilot tested and revised. The survey questionnaire, itself, was distributed primarily through the social media platform, Instagram. This was accomplished by creating an Instagram account page for the survey with a link to the questionnaire

on it, and by following youth accounts. The questionnaire, itself, could be completed on a mobile device, or using a laptop/computer.

Collected data will be stored on secure servers for 5 years and are only accessible by the research team.

## Survey description

The survey began with participants' consent, and then asked for age (13–18) and for self-identity in terms of gender with options, 'boy', 'girl', or 'I do not identify with the gender binary'. The survey included 4 sections each covering one aspect of subjective health (physical health, mental health, life satisfaction, and body image). This article focuses on findings for the indicator of mental health which was defined explicitly for survey participants as a "feeling of well-being that includes believing you are capable of achieving what you hope to achieve, dealing with stresses in life, going to school and working with enjoyment, being productive, and functioning as a part of your community" [20]. Then, participants rated their mental health by selecting one option: 'Excellent', 'Very good', 'Good', 'Fair', or 'Poor'. Guided by existing evidence as to individual and social circumstances that determine adolescent mental health we developed a list of circumstances that participants have been thought to consider while rating their mental health and asked them directly how important each was for this rating via a 4-point scale from 'not important at all' to 'very important'. The main categories of such circumstances included, family (e.g., relationship with parents, family's access to money) [25], peer relationships (e.g., how my peers treat me) [14], personal factors (e.g., future plans, self-acceptance) [25], community involvement (e.g., being part of my community) [26], health behaviors (e.g., exercise routine) and physical health (e.g., physical activities) [27] (see Box 1).

---

### Box 1

We want to know what you thought about when making this rating. For each example given below, please tell us how **important** it was to YOUR rating.

1. My relationships with my parents (ex. loving or challenging, accepting or rejecting, etc.)

2. How my family interacts with each other (ex. feeling connected to my family, family conflict)

3. The way my parents treat me (ex. parental warmth, strictness)

4. My family's access to money

5. How my peers treat me (ex. friendships, bullying/rejection)

6. Comparing myself to my peers

7. Pressure I feel from my peers

8. Having a boyfriend or girlfriend

9. My school performance (ex. grades)

10. Pressures from school (ex. homework)

11. Extracurricular activities (ex. Music or drama clubs, student government, sports)

12. My plans for the future (ex. future opportunities, stresses)

---

13. My involvement in community activities

14. Being part of my community (ex. how I am treated because of my race, religion, sexual orientation, gender, etc.)

15. My self confidence (ex. how confident I feel that I can accomplish things and express myself)

16. Sense of identity (ex. knowing who I am and what I believe in)

17. My ability to cope with things that happen in my life

18. Self-acceptance (ex. how well I like myself)

19. Physical well-being (ex. physical illnesses, how my body feels)

20. A mental health diagnosis

21. My emotional wellbeing (ex. How I deal with different emotions, ongoing effects from bad past experiences)

22. My exercise routine

23. How well I sleep

24. Screen time (ex. amount of time I spend on my phone, computer, video games, etc.)

25. The food I eat

26. The substances I use (ex. alcohol, weed)

## Data analysis

We first compiled descriptive analyses of sex and age distributions as well as the differences in what participants indicated as high- or low-weighted factors in their consideration of SRMH. The average importance of each item for inclusion in rating of SRMH was estimated and compared via student t-tests.

Second, to represent participants' interpretations of meanings of the SRMH measure a factor analysis was conducted. The goal was to determine which factors were most important and to identify groupings of factors that shape the outcomes for everyone and for girls and boys separately. Factor analysis performs the latter by identifying the potential latent factors within the items that participants considered important in rating their mental health. The principal components method was used to extract potential underlying components (latent factors), followed by an orthogonal rotation. The reason for choosing an orthogonal over oblique solution was that the initial extracted components in data were not correlated. In initial exploratory analyses we followed Kaiser Criterion [28] and retained factors with eigenvalues greater than 1, and also Cattell's Scree test [29], which involves an examination of a plot of the eigenvalues for breaks or discontinuities, to determine the number of components to retain. In interpreting the component structure, an item was said to load onto a given component if factor loading was greater than or equal to 0.50 and lower than 0.30 to the other component. The difference between loadings onto the two components also needed to be greater than 0.2. To establish *a priori* decision guidelines, components that did not meet the above criteria were excluded from analysis. The robustness of this exploratory factor analysis was assessed by estimating diagnostic measures of sampling adequacy (the Kaiser–Meyer–Olkin measure) and by

Bartlett's test of sphericity (to determine whether the correlations between the variables, examined simultaneously, do not differ significantly from zero, in other words to evaluate if the two retained components together are sufficient to explain correlations between items). To identify possible differences in extracted domains across sex and age groups, we repeated the factor analyses in boys and girls and also in the age groups of 13–16 and 17–18 years old separately. Final scores of retained components for each participant were calculated via linear regression models. 95% CI intervals around the average of these scores were calculated by bootstrapping with 1000 iterations and compared between sex and age groups. All analyses used SPSS, version 27.

## Ethical considerations

Ethics approval was obtained from the Queen's University Health Sciences and Affiliated Teaching Hospitals Research Ethics Board. Consent was demonstrated by participants' continuing from the information provided in the pre-amble, to the questionnaire and to submission of it. There were several points at which non-consent could be demonstrated. Those who accessed the online invitation could opt out of participation by not proceeding to the questionnaire. The questionnaire, itself, explained that one could opt out at any point by not submitting the questionnaire. The questionnaire was anonymous and confidentiality was ensured by preventing generation of any link between participants' IP addresses, emails or any online individual presence. Participants could choose to not answer specific questions and opt out at any time. Exiting the browser would remove all prior answers.

A small incentive was used for participants, this was a 1/250 chance to win a $100 gift card upon survey completion. Those who entered the prize draw did so via a separate platform to which they were directed after they exited the questionnaire.

## Results

The 506 girls surveyed were almost twice as likely as were the 216 boys to state that their SRMH was poor (47.8% girls versus 27.8% boys) (Table 1).

Those identifying as non-binary (n = 43) were too small in number for sub-analysis but are included when data for all participants are analysed. All participants contemplated similar circumstances in making their determination of SRMH (Table 2). A few notable sex/gender differences did, however, emerge. Boys were more likely to take into account sense of identity and self-confidence, physical well-being, exercise, foods eaten and screen time, while girls paid more attention to having a boyfriend or girlfriend, comparisons with peers, and school pressure and performance than did boys.

With factor analysis five components emerged for the group as a whole (Table 3). The first, which we will refer to as resilience, combined 'self confidence', 'sense of identity', 'ability to cope' and 'self-acceptance'. The second component, behavior/community, merged 'exercise', 'screentime/use', 'sleep patterns', 'substance use', and 'type of food eaten' along with community engagement. Third was family (including relationship with and treatment by parents, family interactions, and sense of family financial resources) and fourth was relationships with peers. The fifth component is one we will call future vision and included items such as 'future plans' and 'performance at school'.

Stratification by sex or age prior to factor analyses yielded similar component to those found for the whole sample (Table 4).

While items making up these components did not change when the whole group was subdivided by age (13–16, 17–18) (S1 Table) a few sex/gender differences surfaced. Overall, boys and girls considered similar components when rating their mental health and there were no

**Table 1. Frequency distributions of self-reported health outcomes in boys, girls and non-binary participants; n (column percentage).**

|  | Boy | Girl | Non Binary | P. value* |
|---|---|---|---|---|
| Age |  |  |  | ANOVA |
| Mean (SD) | 16.4 (1.5) | 16.1 (1.4) | 16.1 (1.3) | 0.045 |
| **Self-Reported Mental Health** |  |  |  | <0.001 |
| Good | 156 (72.2) | 264 (52.2) | 11 (25.6) |  |
| Bad | 60 (27.8) | 242 (47.8) | 32 (74.4) |  |
| **Self-Reported Health** |  |  |  | <0.001 |
| Good | 183 (85.1) | 389 (75.5) | 24 (55.8) |  |
| Bad | 32 (14.9) | 113 (22.5) | 19 (44.2) |  |
| **Body Image** |  |  |  | <0.001 |
| Good | 145 (67.4) | 254 (50.2) | 18 (41.9) |  |
| Bad | 70 (32.8) | 252 (49.8) | 25 (58.1) |  |
| **Life Satisfaction High** | 41 (20.3) | 51 (11.1) | 1 (2.4) | <0.001 |
| Average | 73 (36.1) | 165 (35.6) | 10 (23.8) |  |
| Low | 88 (43.6) | 247 (53.3) | 31 (73.8) |  |

*p values are from Chi-square except for age

significant differences between the importance of these, as demonstrated by comparable average scores of extracted components (Fig 1).

In absolute terms, however, resilience was more important (showing higher score) for girls whereas family, peer, and future were more important for boys. We also observed minor variations by sex in the items that loaded onto components. Screen time was part of the domain, peers, for girls but behaviour for boys, while treatment by peers contributed to the peers domain only among girls. Plans for the future emerged as a component of 'future' for girls but not boys. Among boys, only, being part of a community contributed to the 'community' domain, a mental health diagnosis was an element of 'family', and sleep was a component of resilience. All five-component solutions seemed sufficient to explain correlations between items (p. values for the Barlett's test of sphericity <0.001) and larger than 0.60 estimates of the Kaiser–Meyer–Olkin test suggested adequate sample size—both evidence of the robustness of the models. No differences in the perception of the importance of extracted components were observed between age groups and comparing average scores did not show any pattern (Fig 2)

## Discussion

We examined whether girls and boys are answering the same question when they rate their mental health, that is, does this measure mean the same thing to male and female groups of participants. Underlying and shaping the research question was strong and consistent evidence of a paradox—that girls generally report lower SRMH but other broad markers of well-being such as academic achievement, resilience, less risk-taking behavior and lower rates of suicide all favor girls [11, 30]. Of further importance is methodologic accuracy; for survey data to be trustworthy each respondent must be independent. If clustering based on group characteristics (such as being a girl or a boy) shapes responses, these clusters must be identified and multilevel analyses performed.

Consistent with previous findings female participants reported poorer mental health than did boys, while those whose gender identity did not fit either of these categories reported the poorest SRMH [2, 14, 31]. These ratings did not, however, arise substantially from girls reflecting upon more or different life circumstances than boys. Instead, both boys and girls (and

**Table 2. Average ratings of importance of characteristics when considering SRMH (range 1 to 4).**

| | | Characteristics considered in rating mental health (SRMH) | | | |
|---|---|---|---|---|---|
| Characteristics | Average overall | Average (Boys) | Average (Girls) | Average (Good SRMH) | Average (Bad SRMH) |
| Relationship with Parents[†] | 3.14 | 3.11 | 3.14 | 3.37 | 2.86 |
| How treated by parents[†] | 3.14 | 3.10 | 3.16 | 3.29 | 2.94 |
| Interactions within family[†] | 2.97 | 2.97 | 3.00 | 3.19 | 2.69 |
| Family access to money[†] | 2.56 | 2.67 | 2.52 | 2.72 | 2.36 |
| Comparing to peers*[†] | 2.95 | 2.75 | 3.03 | 2.82 | 3.12 |
| Pressure from peers | 2.49 | 2.39 | 2.53 | 2.45 | 2.55 |
| Having a boy/girl friend* | 2.07 | 2.31 | 1.98 | 2.08 | 2.07 |
| How treated by peers[†] | 2.89 | 2.89 | 2.90 | 2.96 | 2.81 |
| Plans for future[†] | 2.48 | 3.45 | 3.51 | 3.54 | 3.41 |
| Extracurricular activities[†] | 2.70 | 2.83 | 2.67 | 2.91 | 2.42 |
| School Pressure* | 3.20 | 3.04 | 3.28 | 3.16 | 3.26 |
| School performance* | 3.38 | 3.17 | 3.47 | 3.41 | 3.34 |
| Being part of a community[†] | 2.54 | 2.64 | 2.47 | 2.71 | 2.32 |
| Involvement in community*[†] | 2.24 | 2.37 | 2.20 | 2.48 | 1.91 |
| Self_acceptance[†] | 3.30 | 3.27 | 3.31 | 3.36 | 3.22 |
| Ability to cope[†] | 3.28 | 3.27 | 3.27 | 3.33 | 3.22 |
| Sense of identity*[†] | 2.99 | 3.15 | 2.92 | 3.10 | 2.85 |
| Self confidence[†] * | 2.27 | 3.32 | 3.26 | 3.37 | 3.14 |
| Emotional wellbeing | 3.26 | 3.26 | 3.24 | 3.31 | 3.19 |
| A mental health diagnosis | 2.58 | 2.59 | 2.54 | 2.54 | 2.62 |
| Physical wellbeing*[†] | 2.91 | 3.09 | 2.87 | 3.12 | 2.63 |
| Exercise routine*[†] | 2.56 | 2.82 | 2.49 | 2.88 | 2.13 |
| Sleeping well[†] | 2.93 | 2.98 | 2.93 | 3.09 | 2.72 |
| Substance use*[†] | 1.95 | 2.12 | 1.91 | 2.04 | 1.83 |
| Food eaten[†] * | 2.81 | 2.86 | 2.78 | 2.95 | 2.62 |
| Screen time*[†] | 2.55 | 2.68 | 2.51 | 2.68 | 2.37 |

*t-test significant at the level of $\alpha = 0.05$ between boys and girls

[†]t-test significant at the level of $\alpha = 0.05$ between good and bad SRMH

younger and older participants) contemplated relationships with family and friends, individual traits that collectively suggest resilience, behaviors, and their futures [32]. Contrary to what we expected, the explanation for girls' poorer mental health did not appear to stem from their displaying a broader view of the circumstances that shape SRMH.

As a sensitivity analysis to better explain gender differences within age groups, we compared average scores of extracted components in four groups defined by sex and age (S1 Fig). Results mirrored the individual interactive contributions of age and sex. For example, younger boys compared to other groups showed higher average scores of the peer component which was due to higher scores in boys (compared to girls) and in 13–16 years compared to 17–18 years.

We hypothesize that although life circumstances that underpin subjective ratings were similar, lived or perceived realities for each domain somehow disadvantage the mental health of girls relative to boys. For example, although all consider the same underlying characteristics, girls' perceptions or realities of relationships with family or friends may be more negative than is the case for boys. In other words, the discrepancy in boys' and girls' mental health seems to be unrelated to sex/gender differences in contemplation of meanings prior to reporting

**Table 3. Factor loadings analysis of SRH_MH, 5-items solution, the whole sample.**

| | Factor 1 Resilience | Factor 2 Behaviours | Factor 3 Family | Factor 4 Peers | Factor 5 Future |
|---|---|---|---|---|---|
| Relationship with Parents | .150 | .108 | .768 | -.065 | .142 |
| How treated by parents | .128 | .061 | .796 | .005 | .113 |
| Interactions within family | .110 | .086 | .721 | -.012 | .162 |
| Family access to money | -.092 | .283 | .525 | .288 | -.128 |
| Comparing to peers* | .030 | -.013 | -.214 | .648 | .307 |
| Pressure from peers | -.135 | .016 | .106 | .735 | .159 |
| Having a boy/girl friend | .107 | .179 | -.044 | .525 | -.157 |
| How treated by peers | .144 | .040 | .329 | .522 | .158 |
| Plans for future | .155 | .273 | .119 | -.041 | .549 |
| Extracurricular activities* | -.075 | .625 | .141 | -.117 | .292 |
| School Pressure | .087 | .033 | .024 | .156 | .714 |
| School performance | .164 | .040 | .173 | .072 | .733 |
| Being part of a community | .117 | .399 | .469 | .139 | -.083 |
| Involvement in community | .007 | .659 | .301 | .031 | -.040 |
| Self_acceptance | .799 | .072 | -.013 | .021 | .100 |
| Ability to cope | .789 | .014 | .115 | .039 | .136 |
| Sense of identity | .544 | .258 | .253 | .059 | .006 |
| Self confidence* | .761 | .179 | -.012 | -.011 | .154 |
| Emotional wellbeing | .758 | .025 | .140 | .035 | .008 |
| A mental health diagnosis | .260 | .184 | .335 | .414 | -.313 |
| Physical wellbeing | .488 | .501 | .255 | .143 | -.011 |
| Exercise routine* | .151 | .742 | .117 | -.034 | .027 |
| Sleeping well | .391 | .484 | .018 | .083 | .145 |
| Substance use | .044 | .407 | .066 | .323 | -.192 |
| Food eaten* | .244 | .567 | .148 | .170 | .044 |
| Screen time | .090 | .515 | -.126 | .188 | .194 |
| Average score with 95%CI (via linear regression) | 0.75 (0.70–0.80) | 0.83 (0.78–0.88) | 0.79 (0.75–0.86) | 0.84 (0.78–0.89) | 0.86 (0.80–0.92) |

KMO measure for sampling adequacy = 0.842

p. value for Barlett's test of sphericity = 0.000

Total percentage of variance explained by five components: 52%

SRMH. Girls' poorer SRMH may, instead, reflect individually or socially mediated inequities in lived experiences and how these work their way 'under the skin'.

There is, however, a second possible hypothesis. The definition of mental health we presented to participants was: "a feeling of well-being that includes believing you are capable of achieving what you hope to achieve, dealing with stresses in life, going to school and working with enjoyment, being productive, and functioning as a part of your community". There is some evidence that adversity is not only a source of poorer mental health but also a driver of resilience and success in academic and work arenas [24]. When youth were questioned about responses to a resilience scale some noted that their sense of optimism and drive to always do better made them avoid rating their current function too highly [24]. Other studies have consistently demonstrated that girls' resilience (as demonstrated by attributes such as self-control, empathy, optimism, self-efficacy) exceeds boys [33]. Entwined with these attributes may be greater introspection and a cautiousness about overstating that all is well. Perceiving or

**Table 4. Factor loadings analysis of SRHMH, 5-items solution, sex stratified.**

| | Girls | | | | | Boys | | | | |
|---|---|---|---|---|---|---|---|---|---|---|
| | Factor 1 Resilience | Factor 2 Behaviours | Factor 3 Family | Factor 4 Peers | Factor 5 Future | Factor 1 Resilience | Factor 2 Behaviours | Factor 3 Family | Factor 4 Peers | Factor 5 Future |
| Relationship with Parents | .149 | .090 | .772 | -.083 | .039 | .200 | .202 | .711 | -.092 | .296 |
| How treated by parents | .109 | .001 | .774 | -.003 | .020 | .115 | .242 | .793 | -.096 | .310 |
| Interactions within family | .082 | .055 | .717 | -.062 | .151 | .176 | .225 | .694 | -.005 | .245 |
| Family access to money | -.134 | .281 | .499 | .275 | -.005 | .079 | .175 | .711 | .273 | -.136 |
| Comparing to peers* | .018 | -.033 | -.315 | .547 | .220 | -.057 | .047 | -.079 | .776 | .302 |
| Pressure from peers | -.177 | -.012 | .066 | .710 | .130 | -.182 | -.018 | .192 | .690 | .289 |
| Having a boy/girl friend | .089 | .092 | -.066 | .507 | -.068 | .263 | .044 | .092 | .659 | -.120 |
| How treated by peers | .121 | -.007 | .313 | .551 | .145 | .129 | -.032 | .420 | .377 | .338 |
| Plans for future | .116 | .288 | .109 | -.136 | .547 | .436 | .077 | .141 | .099 | .413 |
| Extracurricular activities* | -.118 | .656 | .082 | -.143 | .188 | .102 | .527 | .108 | .024 | .563 |
| School Pressure | .043 | .007 | .004 | .163 | .751 | .262 | .012 | .024 | .150 | .711 |
| School performance | .147 | -.048 | .109 | .062 | .752 | .291 | .198 | .174 | .088 | .663 |
| Being part of a community | .068 | .380 | .476 | .229 | -.018 | .118 | .598 | .365 | .000 | .039 |
| Involvement in community | -.035 | .644 | .282 | .025 | -.035 | .025 | .827 | .217 | .098 | .137 |
| Self_acceptance | .838 | .059 | -.053 | -.026 | .033 | .669 | .107 | .072 | .100 | .136 |
| Ability to cope | .814 | -.060 | .076 | .007 | .134 | .630 | .180 | .215 | .068 | .207 |
| Sense of identity | .521 | .218 | .259 | .112 | .085 | .566 | .379 | .170 | -.048 | .136 |
| Self confidence* | .791 | .120 | -.018 | -.006 | .147 | .747 | .173 | .001 | -.035 | .085 |
| Emotional wellbeing | .794 | .047 | .116 | .045 | .015 | .621 | -.125 | .325 | .063 | .022 |
| A mental health diagnosis | .259 | .179 | .255 | .479 | -.242 | .201 | .206 | .625 | .281 | -.186 |
| Physical wellbeing | .513 | .437 | .262 | .245 | -.087 | .559 | .432 | .238 | .122 | .271 |
| Exercise routine* | .150 | .727 | .105 | .030 | -.021 | .319 | .661 | .151 | .058 | .029 |
| Sleeping well | .399 | .469 | .027 | .051 | .044 | .557 | .343 | .045 | .294 | .134 |
| Substance use | .035 | .384 | -.043 | .322 | -.124 | .176 | .291 | .389 | .374 | -.243 |
| Food eaten* | .277 | .530 | .134 | .203 | -.033 | .295 | .620 | .147 | .256 | .067 |
| Screen time | .136 | .521 | -.133 | .035 | .300 | .139 | .165 | .026 | .555 | -.011 |
| Average score with 95%CI (via linear regression) | 0.84 (0.78–0.90) | 0.83 (0.79–0.88) | 0.70 (0.65–0.75) | 0.70 (0.65–0.74) | 0.79 (0.74–0.84) | 0.75 (0.70–0.80) | 0.83 (0.78–0.88) | 0.80 (0.75–0.86) | 0.84 (0.78–0.89) | 0.86 (0.80–0.92) |

KMO measure for sampling adequacy for girls = 0.808; for boys = 0.836

p. value for Barlett's test of sphericity for both girls and boys <0.001

Total percentage of variance explained by five components for girls = 50%; for boys = 58%

reporting less than perfect SRMH may actually propel girls' greater resilience and academic accomplishment. The direction of this possible association is difficult to determine. Does poorer SRMH lead to greater resilience, does greater resilience colour the reporting of subjective mental health, or is the relationship bidirectional? Of course, even though adversity may be a necessary precursor of resilience this does not justify allowing children or youth to suffer when that suffering can be alleviated. At the same time the strength that arises from facing adversity and navigating it suggests that trying to shelter youth from distressing circumstances may harm as well as help by diminishing self-mastery, self-efficacy, or other components of resilience [34, 35].

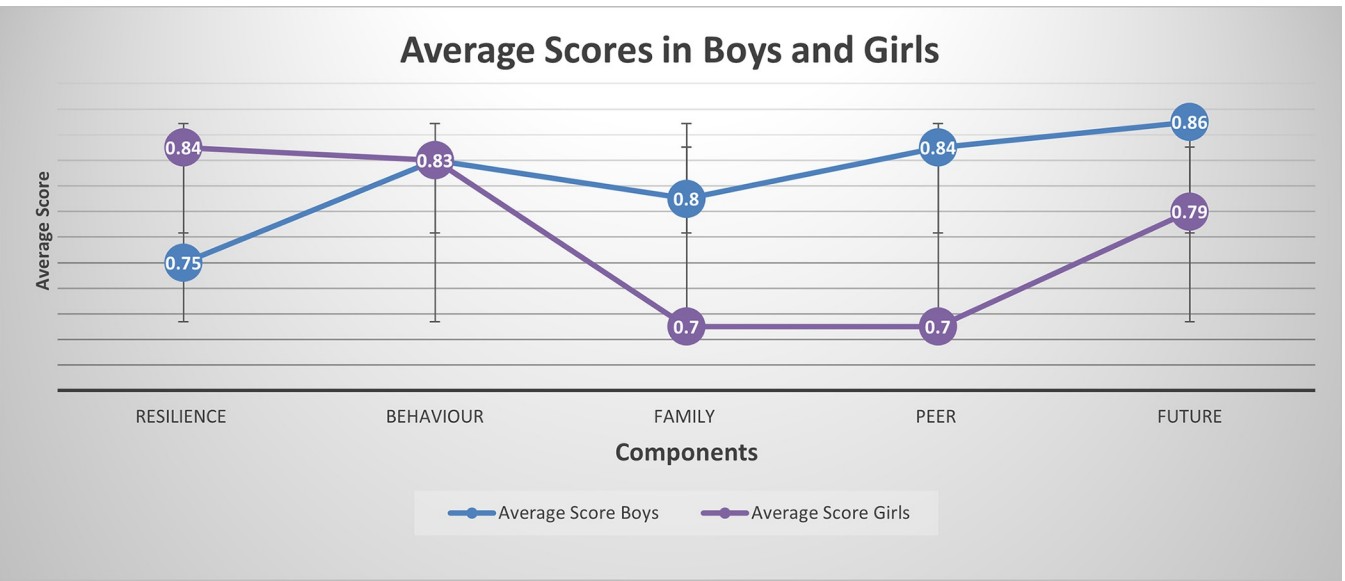

**Fig 1. Average scores of extracted components for boys and girls.**

## Limitations

Social media recruiting targeted Canadian youth, and despite the large sample size, we cannot assume external validity. The minority of Canadian youth who lack access to social media would have been excluded. As with most surveys, girls were twice as likely as boys to participate. Cross-sectional data can identify associations but not directionality or causation. We listed characteristics known to be of relevance and asked participants whether and to what degree they considered each before rating their mental health. There may have been over-reporting about any of these, particularly among less introspective respondents who, in

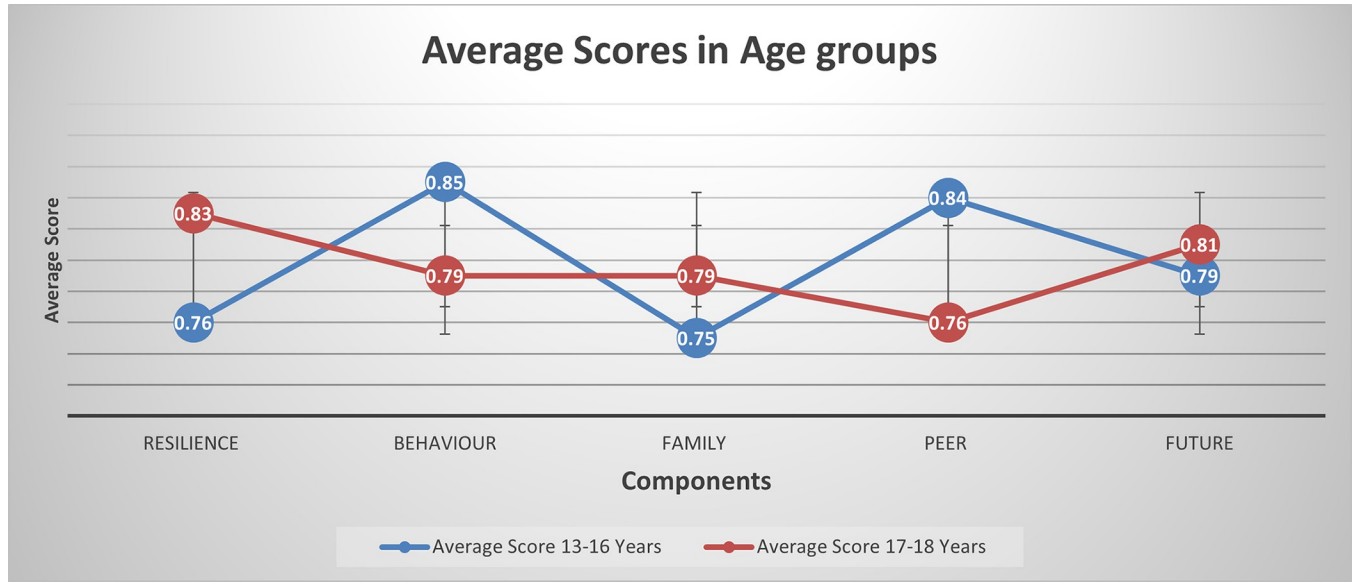

**Fig 2. Average scores of extracted components for 13–16 years and 17–18 years.**

retrospect, may have thought each characteristic was a reasonable one to have considered. As boys are generally thought to be less introspective than girls it is possible that they (boys) disproportionately over-reported weightings of the various characteristics listed. Finally, qualitative research would be needed to identify characteristics we did not list as options.

## Conclusions

In summary, we examined whether the paradox of girls' poorer self-reported mental health might arise from differences in what boys and girls consider as the meanings behind the measure, SRMH. In keeping with others' findings, girls surveyed had much poorer SRMH than did boys. However, both groups considered relatively similar lived circumstances in making their rating. This suggests that SRMH is a valid measure across groups defined by sex. It would appear either that girls realities are more challenging, or that they perceive them as more challenging, perhaps to drive the greater achievement and success they attain. Although our findings allow for hypotheses, empirical explanations for girls' poorer SRMH but greater well-being remain a paradox. To deepen understanding of this paradox further studies of diverse populations and analysis of representative data is warranted. Such studies should include the collection of qualitative as well as quantitative data.

## Supporting information

**S1 Fig. Average scores of extracted components in the four age and sex groups.**
(TIF)

**S1 Table. Factor loadings analysis of SRH_MH, 5-items solution.** Stratified by age groups.
(DOCX)

## Author Contributions

**Conceptualization:** Susan P. Phillips, Fiona Costello, Naomi Gazendam.

**Data curation:** Susan P. Phillips, Fiona Costello, Naomi Gazendam.

**Formal analysis:** Susan P. Phillips, Fiona Costello, Afshin Vafaei.

**Funding acquisition:** Susan P. Phillips.

**Methodology:** Susan P. Phillips, Afshin Vafaei.

**Project administration:** Susan P. Phillips.

**Writing – original draft:** Susan P. Phillips, Fiona Costello, Naomi Gazendam, Afshin Vafaei.

**Writing – review & editing:** Susan P. Phillips, Fiona Costello, Naomi Gazendam, Afshin Vafaei.

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
