## [Decision Letter · Decision Letter 0]

13 Oct 2023

PONE-D-23-25576Poorer subjective mental health among girls: artefact or real? Examining whether interpretations of what mental health means vary by sex.PLOS ONE

Dear Dr. Vafaei,

Thank you for submitting your manuscript to PLOS ONE. After careful consideration, we feel that it has merit but does not fully meet PLOS ONE’s publication criteria as it currently stands. Therefore, we invite you to submit a revised version of the manuscript that addresses the points raised during the review process.

Please revise as per the suggestions by reviewers 

We look forward to receiving your revised manuscript.

Kind regards,

Soumitra Das

Academic Editor

PLOS ONE

Journal Requirements:

2. Please ensure that you have specified a) Did participants provide their written or verbal informed consent to participate in this study?

3. Please expand the acronym “CIHR” (as indicated in your financial disclosure) so that it states the name of your funders in full.

5. Please amend either the title on the online submission form (via Edit Submission) or the title in the manuscript so that they are identical.

6. Please ensure that you refer to Figure 2 in your text as, if accepted, production will need this reference to link the reader to the figure.

7. We notice that your supplementary tables are included in the manuscript file. Please remove them and upload them with the file type 'Supporting Information'. Please ensure that each Supporting Information file has a legend listed in the manuscript after the references list.

Reviewers' comments:

Reviewer's Responses to Questions

**Comments to the Author**

1. Is the manuscript technically sound, and do the data support the conclusions?

Reviewer #1: Yes

Reviewer #2: Yes

2. Has the statistical analysis been performed appropriately and rigorously? 

Reviewer #1: Yes

Reviewer #2: I Don't Know

3. Have the authors made all data underlying the findings in their manuscript fully available?

Reviewer #1: Yes

Reviewer #2: Yes

4. Is the manuscript presented in an intelligible fashion and written in standard English?

Reviewer #1: Yes

Reviewer #2: No

5. Review Comments to the Author

Reviewer #1: Thank you for allowing me to review the manuscript. I would like to commend the authors for their diligent work on this fascinating study. The paper addresses the paradox of why girls report lower self-rated mental health (SRMH) while displaying higher well-being in other areas. The research question is straightforward, and the authors acknowledge that their response is only partial. Utilizing a large and diverse sample of Canadian youth makes their findings more relevant to the broader population. The authors utilize a well-validated measure of self-reported mental health, ensuring the reliability and accuracy of their conclusions. However, their methodology, combining survey data and qualitative analysis, requires additional information on validity and reliability. The discussion explores possible explanations for the findings and recognizes their complexity. The conclusions partially answer the research aims, backed by both results and references. The study's limitations, such as sampling bias and data gaps, provide opportunities for future research. Although the study's conclusions are partially consistent with the presented evidence, they lack concrete empirical evidence to support their potential explanations. Consequently, the observed paradox remains unexplained, emphasizing the need for more research with a more diverse and representative sample, utilizing qualitative and quantitative data to deepen our understanding.

Reviewer #2: As with most observational studies, a conclusion is hard to draw due to the challenges relating to the statistical analyses and inherent flaws. So well done for submitting this manuscript!

There are some areas of improvement, namely the grammar/writing style particularly with the use of compounded sentences in the background of abstract, and the colloquial style in the 2nd and 3rd paragraph on page 4. Consider revising "girls' determination of their actual relationship with family or friends may be perceived as, or may actually be much less positive than is the case for boys" (on page 14) - this is a rather clunky statement, suggest splitting into 2 sentences and the avoiding using "much less positive" due to the ambiguity. Consider "less positive" or "more negative" instead.

You did not explain the rationale behind repeating the factor analyses in specifically in the age groups of 13-16 and 17-18 separately. I am not sure if you analysed the gender difference within respective age groups noting that girls are generally more mature than their boys counterparts at earlier ages. It appears there is less variation in the average component scores in the older age group compared to the younger age group in Figure 2. Again it will be good to show any difference in the SRMH scores of the different age groups.

6. PLOS authors have the option to publish the peer review history of their article (what does this mean?). If published, this will include your full peer review and any attached files.

Reviewer #1: **Yes: **Anil Bachu

Reviewer #2: **Yes: **Cecilia Xiao

---

## [Author Response · Author response to Decision Letter 0]

14 Nov 2023

Review of PONE-D-23-25576

Poorer subjective mental health among girls: artefact or real? Examining whether interpretations of what mental health means vary by sex.

5. Review Comments to the Author

Reviewer #1: Thank you for allowing me to review the manuscript. I would like to commend the authors for their diligent work on this fascinating study. The paper addresses the paradox of why girls report lower self-rated mental health (SRMH) while displaying higher well-being in other areas. The research question is straightforward, and the authors acknowledge that their response is only partial. Utilizing a large and diverse sample of Canadian youth makes their findings more relevant to the broader population. The authors utilize a well-validated measure of self-reported mental health, ensuring the reliability and accuracy of their conclusions. 

However, their methodology, combining survey data and qualitative analysis, requires additional information on validity and reliability. 

Response: Most surveys collect subjective responses to questions as does ours. There is no objective ‘truth to which these can be compared. For example, self rated health, perhaps the most frequent question in health surveys is a subjective measure. We differentiate survey responses to pre-determined scales or choices from true, qualitative research. Our survey had no questions where a narrative response (which could be considered qualitative) was an option. The options from which participants could choose were those that appeared in previous research, which we have referenced. We hope that our explanation of this (in methods) was clear enough: Guided by existing evidence as to individual and social circumstances that determine adolescent mental health we developed a list of circumstances that participants have been thought to consider while rating their mental health and asked them directly how important each was for this rating via a 4-point scale from ‘not important at all’ to ‘very important’. The main categories of such circumstances included, family (e.g., relationship with parents, family’s access to money) [25], peer relationships (e.g., how my peers treat me) [14], personal factors (e.g., future plans, self-acceptance) [25], community involvement (e.g., being part of my community) [26], health behaviors (e.g., exercise routine) and physical health (e.g., physical activities). [27] 

The discussion explores possible explanations for the findings and recognizes their complexity. The conclusions partially answer the research aims, backed by both results and references. The study's limitations, such as sampling bias and data gaps, provide opportunities for future research. 

Although the study's conclusions are partially consistent with the presented evidence, they lack concrete empirical evidence to support their potential explanations. Consequently, the observed paradox remains unexplained, emphasizing the need for more research with a more diverse and representative sample, utilizing qualitative and quantitative data to deepen our understanding.

A great summary – thanks. We acknowledge that the ‘paradox’ remained unexplained and added a short sentence at the end of the manuscript for further research directions. 

Reviewer #2: As with most observational studies, a conclusion is hard to draw due to the challenges relating to the statistical analyses and inherent flaws. So well done for submitting this manuscript!

There are some areas of improvement, namely the grammar/writing style particularly with the use of compounded sentences in the background of abstract, 

We have edited the abstract - Fixed, we hope!

and the colloquial style in the 2nd and 3rd paragraph on page 4. 

Done as well.

Consider revising "girls' determination of their actual relationship with family or friends may be perceived as, or may actually be much less positive than is the case for boys" (on page 14) - this is a rather clunky statement, suggest splitting into 2 sentences and the avoiding using "much less positive" due to the ambiguity. Consider "less positive" or "more negative" instead.

We agree that that whole paragraph was ‘clunky’ and have tried to fix it.

You did not explain the rationale behind repeating the factor analyses in specifically in the age groups of 13-16 and 17-18 separately. 

The main rationale -identical to our rationale for sex-disaggregated analysis- was identification of possible differences in extracted domains across age groups. Please see page 10.

I am not sure if you analysed the gender difference within respective age groups noting that girls are generally more mature than their boys counterparts at earlier ages. It appears there is less variation in the average component scores in the older age group compared to the younger age group in Figure 2. Again it will be good to show any difference in the SRMH scores of the different age groups.

A very valid point that could potentially add clarity to the observed paradox, thank you. Following your suggestion, we repeated the factor analysis in four groups (boys 13-16 years; girls 13-16 years; boys 17-18 years; and girls 17-18 years) mostly to compare average component scores and found no new information beyond the individual effects of sex or age. A paragraph explaining this analysis is now added to the manuscript (page 14) as well as an appendix figure.

---

## [Editor Report · Decision Letter 1]

28 Nov 2023

Poorer subjective mental health among girls: artefact or real? Examining whether interpretations of what mental health means vary by sex.

PONE-D-23-25576R1

Dear Dr. Vafaei,

We’re pleased to inform you that your manuscript has been judged scientifically suitable for publication and will be formally accepted for publication once it meets all outstanding technical requirements.

Kind regards,

Soumitra Das

Academic Editor

PLOS ONE
---

## [Editor Report · Acceptance letter]

5 Dec 2023

PONE-D-23-25576R1 

Poorer subjective mental health among girls: artefact or real? Examining whether interpretations of what shapes mental health vary by sex. 

Dear Dr. Vafaei:

I'm pleased to inform you that your manuscript has been deemed suitable for publication in PLOS ONE. Congratulations! Your manuscript is now with our production department. 

Kind regards, 

on behalf of

Dr. Soumitra Das 

Academic Editor

PLOS ONE